# Effects of problem-based learning modules within blended learning courses in medical statistics – A randomized controlled pilot study

**Zoran Bukumiric**[1]*, **Aleksandra Ilic**[2], **Mirjana Pajcin**[2], **Dragana Srebro**[1], **Sasa Milicevic**[2], **Dragan Spaic**[3], **Nenad Markovic**[3], **Aleksandar Corac**[2]

**1** University of Belgrade, Faculty of Medicine, Belgrade, Serbia, **2** Faculty of Medicine, University of Pristina in Kosovska Mitrovica, Kosovska Mitrovica, Serbia, **3** Faculty of Medicine, University of East Sarajevo, Foca, The Republic of Srpska, Bosnia and Herzegovina

* zoran.bukumiric@med.bg.ac.rs

**Data Availability Statement:** All relevant data are within the manuscript and its Supporting Information files.

## Abstract

Problem-based learning (PBL) allows students to learn medical statistics through problem solving experience. The aim of this study was to assess the efficiency of PBL modules implemented in the blended learning courses in medical statistics through knowledge outcomes and student satisfaction. The pilot study was designed as a randomized controlled trial that included 53 medical students who had completed all course activities. The students were randomized in two groups: the group with access to PBL modules within the blended learning course (hPBL group) and the group without access to PBL modules–only blended learning course (BL group). There were no significant differences between the groups concerning socio-demographic characteristics, previous academic success and modality of access to course materials. Students from hPBL group had a significantly higher problem solving score (p = 0.012; effect size 0.69) and the total medical statistics score (p = 0,046; effect size 0.57). Multivariate regression analysis with problem solving as an outcome variable showed that problem solving was associated with being in hPBL group (p = 0.010) and having higher grade point average (p = 0.037). Multivariate regression analysis with the medical statistics score as an outcome variable showed the association between a higher score on medical statistics with access to PBL modules (p = 0.045) and a higher grade point average (p = 0.021). All students in hPBL group (100.0%) considered PBL modules useful for learning medical statistics. PBL modules can be easily implemented in the existing courses within medical statistics using the Moodle platform, they have high applicability and can complement, but not replace other forms of teaching. These modules were shown to be efficient in learning, to be well accepted among students and to be a potential missing link between teaching and learning medical statistics. The authors of this study are planning to create PBL modules for advanced courses in medical statistics and to conduct this study on other universities with a more representative study sample, with the aim to overcome the limitations of the existing study and confirm its results.

**Funding:** The author(s) received no specific funding for this work.

**Competing interests:** The authors have declared that no competing interests exist.

## Introduction

Medical doctors can be exceptional in their fields even if they do not know medical statistics, but they will be better if they do [1]. The study of Swift et al [2] showed that medical doctors considered medical statistics useful for "accessing clinical guidelines and evidence summaries, explaining risk levels to patients, assessing medical marketing and advertising material, interpreting the results of a screening test, reading research publications for general professional interest, and using research publications to explore non-standard treatment and management options". Future physicians also thought that there was a need for a practical application of knowledge in medical statistics, not only its' theoretical basis [3]. Lack of knowledge in medical statistics can lead to misinterpretation of clinical findings [4]. Statistical softwares, widely available now, enable an easy and comfortable analysis, but mistakes can be made when choosing the appropriate statistical test or assumptions for its' application [5]. Medical students state that learning medical statistics through real life problems and the process of drawing conclusions can be more productive than traditional learning and knowledge assessment [6].

The process of education in medical sciences is most commonly based on traditional classroom lectures (face-to-face, lecture-based). In the past decade, there has been an increasing number of studies aiming to test, improve and introduce other forms of teaching, such as e-learning, blended learning, problem-based learning, team-based learning and flipped classroom [7–9].

A need to modify traditional classroom learning became a focal topic during the COVID-19 pandemic when teaching activities at universities worldwide were forced to shift to different types of online learning. It appears that in the future this challenge will be permanently changing the methods of physicians' education [10]. The findings of new possibilities to transfer knowledge and skills through online learning modules and its' constant improvement are receiving almost universal attention. In accordance with this, the implementation of problem-based learning in the online environment has shown similar success among students compared to in-person problem-based learning [11].

Problem-based learning enables students to learn through problem-solving experience [12]. During the learning process students' main focus is on understanding and solving problems, rather than memorizing facts. Students develop critical thinking and clinical reasoning in concrete medical situation, which is very important for physicians' daily practice [13]. There are positive experiences of combining problem-based learning (PBL) with lecture-based learning (LBL) and blended learning in education of future health care professionals [14, 15]. A combined model of PBL + LBL was shown to be efficient in increasing the knowledge score, skills score and students' satisfaction. This hybrid approach in learning has been increasingly used in Chinese medical faculties recently [16].

To the best of our knowledge, there are no previous studies on problem-based blended learning method for teaching medical statistics.

The aim of this study was to evaluate the effectiveness of implemented problem-based modules within blended learning courses in medical statistics through the outcomes of knowledge and student satisfaction. We created problem-based modules in medical statistics, based on actual problems which contained all of the steps in statistical analysis (defining the problem, choosing and applying adequate statistical tests, interpreting the results and drawing conclusions) and implemented them within the blended learning course.

## Materials and methods

The study was designed as a randomized controlled trial that included third-year medical students at the Faculty of Medicine, University of Pristina, Kosovska Mitrovica. The final analysis

included 53 students who had completed all course activities out of 62 students who had been initially included in the study. Students were randomized in two groups: the group with access to problem-based modules within the blended learning curriculum (hybrid problem-based learning group–hPBL group) and the group with no access to problem-based learning modules–only blended learning course (blended learning group–BL group). The study began on October 1[st], 2019, at the beginning of the academic year, and was completed at the end of the academic year (September 30[th], 2020). As the research took part during the entire academic year, a part of the study was conducted during the COVID-19 pandemic. Classes in medical statistics and informatics were organized as a blended learning module.

Problem-based modules were conceptualized as an addition to the existing theoretical and practical curriculum in medical statistics within the blended learning course. PBL modules were created based on the technical solution verified by the board of the Ministry of Education, Science and Technological Development of the Republic of Serbia [17].

Blended learning course in medical statistics and informatics is based on a Moodle platform and contains 15 classes of theoretical lectures, 30 classes of practical exercises and 15 classes of other type, such as online readings or seminars. Total of 70% of the program of this course is comprised of medical statistics and this part of the course contains units on data types, descriptive statistics, confidence interval, probability and probability distributions, hypotheses testing, correlation and linear regression. Practical exercises are done using the statistical software Easy R (EZR) [18]. Students from both groups in our study had access to identical course activities (lectures and exercises), except for the access to the problem-based modules that were available only the students from hPBL group (Table 1). During the course, students receive grades for all existing activities: lectures, exercises, colloquium, seminars, solving problems and final test. Students can see all the points for each activity any time during the course. The maximal number of points is 100 (70 for statistics and 30 for informatics). Students need to obtain the minimum of 51 points to pass the course. Based on the total number of points (51–100) the passing grades students can receive vary from 6 to 10.

Our study examined only the outcomes of medical statistics (70% of the course): problem solving score (5 problems with maximum score of the total of 25 points) and total medical statistics score (theoretical knowledge score, practical exercises score, problem solving score, independent students' assignments score, seminars and colloquium; the maximal total medical statistics score was 70 points). The final grade in the course could not be compared because of

**Table 1.  Activities during medical statistics and informatics course.**

| Timeline | hPBL | BL |
|---|---|---|
| Weekly | Lectures | Lectures |
| Weekly | Practical exercises done using the statistical software | Practical exercises done using the statistical software |
| Weekly | Independent students' assignments (interactive online lectures, Moodle) | Independent students' assignments (interactive online lectures, Moodle) |
| Weekly | Problem-based learning module (Moodle) | – |
| During the course | Seminars | Seminars |
| During the course | Colloquium | Colloquium |
| At the end of the course | Problem solving | Problem solving |
| At the end of the course | Final test | Final test |

the score in medical informatics, since the course contains both medical statistics and medical informatics. An anonymous online questionnaire using the five-point Likert scale (1 point-low satisfaction, 5 points- high satisfaction) was used to assess the students' satisfaction in hPBL group with the PBL modules.

PBL modules were created based on the structure of the steps in statistical analysis (Fig 1). Statistical analysis of the research problem is based on the multiple successive steps which include the following: the definition of the problem and the research question, recognition of the data type, sample type and hypothesis, selection of the adequate statistical test, application of the test, interpretation of the results and conclusion related to the description of the data, statistical conclusion and implications of the results. The PBL modules use guiding questions following the steps of statistical analysis. The guiding questions consisted of: interactive multiple choice or open-ended questions and followed a similar principle to the one Brown et al. applied [19]. Guiding questions changed within each step, based on the type of the statistical problem, number of variables and the sample (examples can be seen online following the link provided in the text below). Students can understand the necessary components for statistical reasoning by answering these questions and learn how to solve the problem.

PBL modules were incorporated in each unit within the blended learning curriculum in Medical statistics and informatics for students in hPBL group. For each unit, students had PBL modules to resolve and to synthesize knowledge from theoretical and practical modules. Each

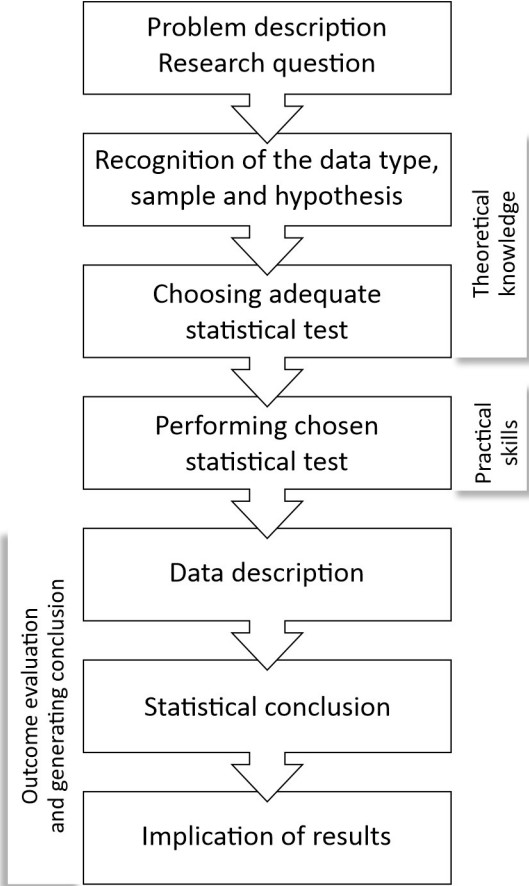

**Fig 1. Basic structure of statistical analysis and the problem-based module.**

PBL module was created and moderated by the teacher (tutor in classical problem-based learning). All the necessary information was given to students during the lectures, while the exercises and materials were given in the blended learning course. Students can resolve the problems alone or in communication with other students within the group (sending messages on Moodle platform, by asking questions in the forum discussion specially designed for these purposes, or by addressing questions to the online moderator- teacher). Students can use all available materials from the blended learning course, as well as other online materials or books during problem solving. The success in each step is evaluated. After completing all the necessary steps, students receive points and depending on the aim of the module, they also receive correct answers. The system supports the possibility of repeating an attempt of solving the problem until the student achieves the minimal necessary knowledge level, or desired knowledge level. PBL module contains meta-cognitive characteristics such as planning, managing and application of the previously adopted knowledge.

Examples of PBL modules used in this study are based on the course Problem based modules in medical statistics and can be accessed via the following link: http://e-ucenje.med.pr.ac.rs/course/view.php?id=232 (username: user; password: user).

## Ethical statement

The study was approved by the Ethical Committee of the Faculty of Medicine, University of Pristina, Kosovska Mitrovica (No. 09–3171). During the first week of the course (the first week of the semester), before the randomization in the groups was performed, students received written and oral explanations of the study, the processes and aims, the modalities of data gathering and data analysis. Students were explained that all the information gathered would be anonymous, that the participation was voluntary and that they could dropout of the study at any point. After this, the students gave an oral consent for their participation in the study, which was then verified in their records. A questionnaire on satisfaction was filled in after the course had been completed, as an online anonymous and non-obligatory questionnaire. All the data on the course outcomes and the data from the questionnaire on the students' satisfaction were gathered by the administrator of the Moodle platform who was the only one with the access to the complete database. The authors of the study only had access to the anonymized database that is provided with the manuscript.

## Statistical analysis

Based on the variable types and normality of distribution, description of the data was shown as number (n) and percentage (%), mean±standard deviation or median (range, minimum- maximum). T-test, Mann-Whitney test, Chi-square test or Fisher's exact test were used to test the hypotheses. The effect size in t-test was examined with Cohen's $d$. Linear regression was used to analyze the learning outcome (problem solving score and total medical statistics score) and its potential predictors. All the variables which were significant in the univariate models at the level of 0.05 were entered in the multivariate regression analyses. Statistical hypotheses were tested at the significance level (alpha) of 0.05.

## Results

There were no statistically significant differences between the students from the hPBL and BL group concerning socio-demographic characteristics, grade point average and the modality of access to the materials within the blended learning course (Table 2).

Students in hPBL group had a significantly higher problem solving score (p = 0.012, effect size 0.69) and total medical statistics score (p = 0.046, effect size 0.57) (Table 3). Students in

**Table 2. Characteristics of students included in the research.**

| Characteristics of students | Total | hPBL | BL | p-value |
|---|---|---|---|---|
| | (n = 53) | (n = 26) | (n = 27) | |
| Age (in years), mean ± sd | 21.4±0.9 | 21.4±1.0 | 21.4±0.9 | 0.934 |
| Sex, n (%) | | | | |
| male | 16 (30.2%) | 8 (30.8%) | 8 (29.6%) | 0.928 |
| female | 37 (69.8%) | 18 (69.2%) | 19 (70.4%) | |
| Grade point average, mean ± sd | 7.7±0.7 | 7.7±0.7 | 7.8±0.8 | 0.611 |
| The grade expected, n (%) | | | | |
| 8 | 4 (8.7%) | 2 (8.7%) | 2 (8.7%) | 0.787 |
| 9 | 13 (28.3%) | 6 (26.1%) | 7 (30.4%) | |
| 10 | 29 (63.0%) | 15 (65.2%) | 14 (60.9%) | |
| Successfully completed the course, n (%) | | | | |
| Before COVID-19 pandemic | 22 (41.5%) | 13 (50.0%) | 9 (33.3%) | 0.218 |
| During the COVID-19 pandemic | 31 (58.5%) | 13 (50.0%) | 18 (66.7%) | |
| Access to fast internet, n (%) | 46 (100.0%) | 23 (100.0%) | 23 (100.0%) | 1.000 |
| Had any online module previously, n (%) | 8 (17.4%) | 3 (13.0%) | 5 (21.7%) | 0.699 |
| Most common time of access to course materials, n (%) | | | | |
| When at faculty or from home | 10 (21.7%) | 4 (17.4%) | 6 (26.1%) | 0.475 |
| Both when at faculty and from home | 36 (78.3%) | 19 (82.6%) | 17 (73.9%) | |
| The most commonly used device for access to the course materials, n (%) | | | | |
| PC or laptop | 40 (87.0%) | 20 (87.0%) | 20 (87.0%) | 1.000 |
| Tablet or smartphone | 6 (13.0%) | 3 (13.0%) | 3 (13.0%) | |
| Self-rated computer skills (1- very poor, 5- very good), median (range) | 4 (1–5) | 4 (3–5) | 4 (1–5) | 0.227 |

hPBL group had a significantly higher score on total satisfaction with the course (median 5, range 4–5) compared to the students in BL group (median 5, range 3–5), (p = 0.012).

Multivariate regression analysis with problem solving as an outcome variable showed that problem solving was associated with being in hPBL group (p = 0.010) and having higher grade point average (p = 0.037) (Table 4 and Fig 2).

Multivariate regression analysis with total medical statistics score as an outcome variable showed that the total medical statistics score was associated with hPBL group (p = 0.045) and higher grade point average (p = 0.021) (Table 5 and Fig 3).

All the students in hPBL group (100.0%) thought that the PBL modules helped them to achieve the desirable knowledge in medical statistics. On the five-point Likert scale (1 –the lowest satisfaction, 5 –the highest satisfaction) median grade for adequacy of the modules, modalities of solving problems, the assistance received in the process of learning medical statistics, the students graded problem solving modules with the median grade 5 (range 4–5). Median on the interest in problem solving modules was 4.5 (range 3–5) (Table 6).

**Table 3. Outcomes among two groups of students.**

| Outcomes | Total | hPBL | BL | p-value |
|---|---|---|---|---|
| | (n = 53) | (n = 26) | (n = 27) | |
| Problem solving score, mean ± sd | 21.8±2.1 | 22.5±1.7 | 21.1±2.3 | 0.012 |
| (range) | (16.1–25.0) | (18.7–25.0) | (16.1–24.7) | |
| Total medical statistics score, mean ± sd | 62.3±4.7 | 63.6±3.8 | 61.0±5.2 | 0.046 |
| (range) | (52.6–68.9) | (55.5–68.9) | (52.6–68.9) | |

**Table 4. Regression models with problem solving score as an outcome variable.**

| Variable | Univariate linear regression | | Multivariate linear regression | |
|---|---|---|---|---|
| | b | p | b | p |
| hPBL vs BL | 1.434 | 0.012 | 1.455 | 0.010 |
| Age | -0.230 | 0.473 | | |
| Sex | -0.353 | 0.583 | | |
| Grade point average | 1.045 | 0.009 | 0.809 | 0.037 |
| The grade expected | 0.680 | 0.156 | | |
| Successfully completed the course during vs before the COVID-19 pandemic | -0.896 | 0.131 | | |
| Had any online module previously | -0.682 | 0.410 | | |
| Most common time of access to course materials | 1.181 | 0.117 | | |
| The most commonly used device for access to the course materials | -2.078 | 0.022 | -1.589 | 0.062 |
| Self-rated computer skills | 0.332 | 0.364 | | |

## Discussion

In this study, it has been shown for the first time that the implementation of problem-based learning into blended learning course in undergraduate medical studies contributes to better learning outcomes in medical statistics. The results of this research indicate that students of the hPBL group had a significantly higher problem solving scores and total medical statistics scores. The presented PBL modules enable active learning of medical statistics by solving actual statistical problems, through conceptual understanding, and they direct students to logical thinking. This direction is in line with recommendations from the Guidelines for Assessment and Instruction in Statistics Education (GAISE) Reports published by the American Statistical Association (ASA) [20]. Also, our results are consistent with the results of a systematic review that relates to the benefits of problem-based learning over traditional learning in biomedical education [21]. Like in our study, medical students had better solving scores and satisfaction [21]. The results of another meta-analysis showed that combined PBL and LBL learning of clinical medicine was significantly superior in achieving higher knowledge and skills scores, as well as learning satisfaction compared to LBL alone [22]. This meta-analysis included studies from China, and its' authors suggested hybrid PBL to be gradually introduced into clinical medical teaching programs [22] since the results from previous meta analysis in China had shown that the competencies of students in medical statistics were insufficient and that they did not have sufficient ability to practically apply their statistical knowledge [23].

Teacher in medical statistics acted as a tutor in our study, he/she was obligated to answer students' questions and to evaluate the outcomes of the problem based learning. In the study of Woltering et al [24] blended problem-based learning included the e-learning module complementary to the classic PBL modules, but without the inclusion of tutors. This study showed an increase in students' motivation, subjective gains in knowledge and overall satisfaction among students in blended program-based learning. There was no significant difference in the successfulness of problem solving, which confirms that the PBL can be successfully implemented in online learning environments. Additionally, in the study which compared traditional classroom classes and online asynchronous PBL, de Jong et al [25] found that the absence of a formal tutor can force students to rely on themselves and teamwork, develop critical thinking, analytical and self-regulation skills.

The results of our study suggest that the modality used to access the course is a significant predictor of the problem solving score (if it is via PC/laptop or Tablet/Smartphone). Students who accessed the course via PC/laptop had better scores, which was expected because the

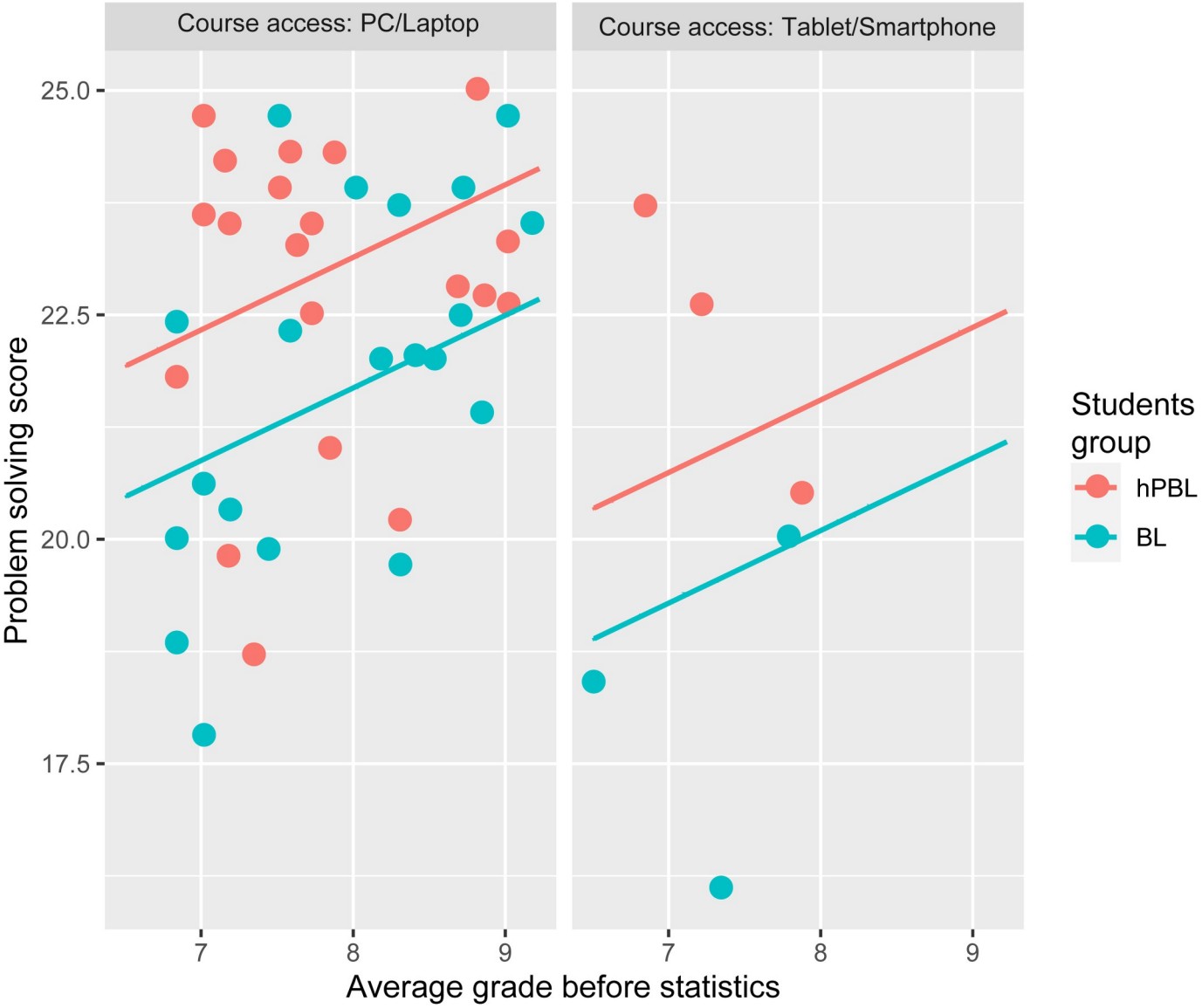

**Fig 2. The relationship between the problem solving score and factors associated with it in multivariate regression model.** *hPBL*–hybrid Problem Based Learning, *BL*–Blended Learning.

problems are solved in the EZR software via PC, and the access via smartphone or tablet does not allow access to this feature. On the other hand, access to theoretical materials may be more comfortable/practical via tablets. However, the method of access is not a statistically significant predictor of total medical statistics score.

Self-rated knowledge through the expected final grade was significantly associated with the total medical statistics score in univariate regression model, which was, most likely, influenced by multiple accesses to PBL modules and students' motivation. The motivation was not directly assessed in our study, but was assessed indirectly, through the expected final grade, as there is a proven positive association between the intrinsic motivation and perceived academic rank [26]. It is expected that highly motivated students would use the advantages of an online platform more and would have higher scores in solving actual problems and higher total medical statistics score.

**Table 5. Regression models with total medical statistics score as an outcome variable.**

| Variable | Univariate linear regression | | Multivariate linear regression | |
|---|---|---|---|---|
| | **b** | **p** | **b** | **p** |
| hPBL vs BL | 2.567 | 0.047 | 2.431 | 0.045 |
| Age | -1.018 | 0.150 | | |
| Sex | 0.603 | 0.674 | | |
| Grade point average | 2.678 | 0.002 | 2.006 | 0.021 |
| The grade expected | 2.643 | 0.010 | 1.849 | 0.059 |
| Successfully completed the course during vs the COVID-19 pandemic | -0.216 | 0.871 | | |
| Had any online module previously | -2.642 | 0.141 | | |
| Most common time of access to course materials | 1.198 | 0.472 | | |
| The most commonly used device for access to the course materials | -3.298 | 0.102 | | |
| Self-rated computer skills | 1.216 | 0.125 | | |

Multivariate regression models showed a significant association between the problem solving score and total medical statistics score with hPBL group and higher grade point average. We expected that students with a higher grade point average have higher problem solving as well as the total medical statistics score, regardless of the study group. Although we did not have a large sample and we did not find a great difference between the means of the two groups, the effect size was moderate. However, the implemented PBL modules can be adapted to teaching in medical statistics, as they offer a modern method of solving practical problems that is appealing to students. Such an approach of directing them through a problem, from its' recognition to the analysis of results, helps students to better understand medical statistics and can help them later during their scientific research work. Depending on the curriculum and concept of teaching medical statistics, PBL can be used as an addition to theoretical and/or practical classes. PBL can be easily implemented in different teaching models (classic, online, blended), which in our case proved to be extremely useful during the COVID-19 pandemic. PBL modules can be used not only during classes for learning or updating materials, but also for practicing assignments and self-assessment of knowledge by students. The PBL is completely independent from the modality of calculation (classical or software) or on platform (can be solved in a classroom, on paper, or using an online study platform). The optimal application of PBL is within online courses because they allow students to access materials when it suits them, as often as they need to (self-regulated learning). Educational reform that combined Moodle with the traditional way of learning medical statistics has achieved good results among students [27]. The meta-analysis showed that the blended learning methods were more efficient than traditional learning in medical sciences [28]. Also, PBL can be used for assessment of knowledge on the exam itself. This is supported by the results from our study, which showed that the time of taking the exam (before or during the COVID-19 pandemic) was not a statistically significant predictor of problem solving score and total medical statistics score. During the COVID-19 pandemic, there were also changes in testing methods [29] and the possibility of introducing the open-book examination [30]. The applied PBL modules can also be used for online assessment of problem solving from medical statistics with the open book model, since in a limited time linking of all information needed for problem solving cannot be compensated by searching on the Internet, books or any study materials.

The potential for the application of these modules is high, in the context of the current level of presence of e-learning methods in medical sciences and especially, due to the increase in digitalization during the COVID-19 outbreak. The PBL can be used for learning, repetition,

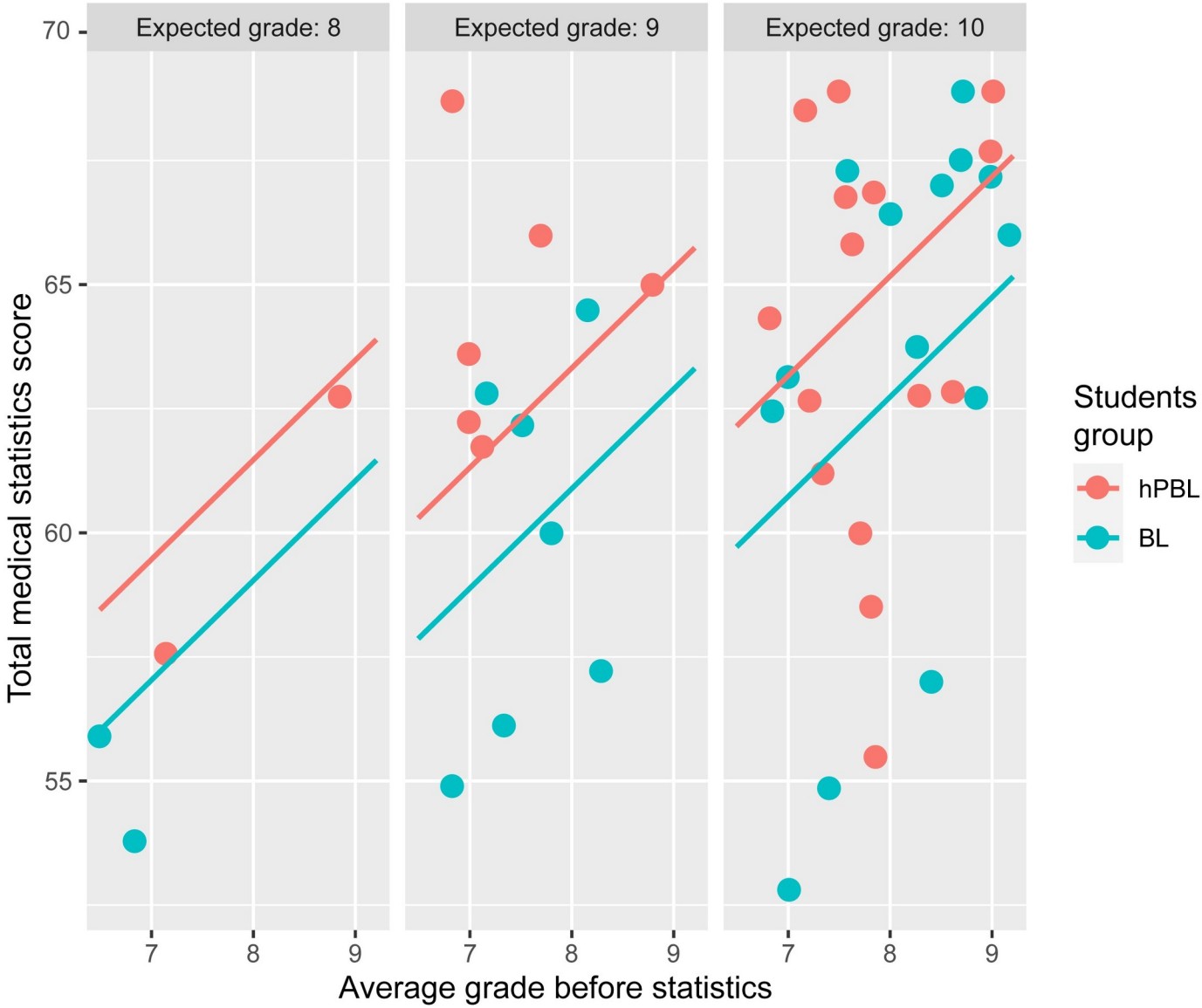

**Fig 3. The relationship between the total medical statistics score and factors associated with it in multivariate linear regression model.** *hPBL*–hybrid Problem Based Learning, *BL*–Blended Learning.

**Table 6. Students' attitudes towards the implemented problem solving modules.**

| Question | median (range) | n (%) highest satisfaction |
|---|---|---|
| Problems are adequate for learning medical statistics | 5 (3–5) | 19 (73.1%) |
| Contents and structure of problem-based modules is interesting | 4.5 (3–5) | 13 (50.0%) |
| I like this modality of solving the actual statistical problems | 5 (3–5) | 19 (73.1%) |
| Step- by- step approach within the module is useful for learning medical statistics | 5 (3–5) | 20 (76.9%) |
| Problem based modules helped me to learn the steps for resolving actual statistical problems | 5 (3–5) | 21 (80.8%) |
| Problem-based modules helped me understand medical statistics | 5 (3–5) | 20 (76.9%) |

exercise, and self-assessment, and finally, for the assessment in the exam. This increases its use-fulness threefold.

The results from the meta- analysis showed that the courses which implemented the PBL were associated with long term knowledge retention, short term retention and application of clinical skills and thinking. Traditional approach is more convenient for short term knowledge retention which does not require further understanding [31]. We did not examine the knowl-edge retention, and it could be included in the further research.

## Limitations

This study has a few possible limitations. Firstly, the study was conducted in only one edu-cational institution, during one study course, with a small number of participants and this should be taken into account when generalizing the results in other courses, study pro-grams or universities. Secondly, the PBL modules include only modules in the basic course of medical statistics. Although this concept and its technical solution can be applied to higher levels of education, there is a need for the assessment of its efficiency, which is our aim for further studies. Another potential limitation of the study we could not control was the fact that the students from the hPBL group could show the modules to the students in BL group. This risk can be minimized through the inclusion of a larger number of partici-pants in the study and through the development of a larger number of the problems for students to solve. The enthusiasm of students and their reaction to the new type of learn-ing, especially during the COVID-19 pandemic, can also affect higher grades among the hPBL group.

## Future research

The authors of this study are planning to create PBL modules for advanced courses in medical statistics and to conduct the study on other universities with a more representative study sam-ple, with the aim to overcome the limitations of the existing study and confirm its results.

## Conclusion

The presented PBL modules can be easily implemented in the existing courses of the medical statistics developed on the Moodle platform, have high applicability and can complement, but not replace other forms of teaching. The PBL modules allow students to associate theory and practice, synthesize the existing knowledge and generate new knowledge and show how medi-cal statistics can be thought through conceptual understanding via directing students through problems using the step-by-step approach. The problem-based modules were shown to be effi-cient in acquiring knowledge, are well accepted among students and can be a missing link in learning and understanding medical statistics.

## Supporting information

**S1 Database.**
(ZIP)

## Acknowledgments

This article is dedicated to our teacher and friend Professor Goran Trajkovic, who passed away prematurely.

## Author Contributions

**Conceptualization:** Zoran Bukumiric.

**Data curation:** Zoran Bukumiric, Aleksandra Ilic, Mirjana Pajcin.

**Formal analysis:** Zoran Bukumiric.

**Investigation:** Zoran Bukumiric.

**Methodology:** Zoran Bukumiric, Mirjana Pajcin, Dragana Srebro.

**Project administration:** Zoran Bukumiric, Aleksandar Corac.

**Resources:** Zoran Bukumiric, Sasa Milicevic, Dragan Spaic, Nenad Markovic, Aleksandar Corac.

**Software:** Zoran Bukumiric, Aleksandra Ilic, Mirjana Pajcin.

**Supervision:** Zoran Bukumiric, Dragana Srebro, Sasa Milicevic, Dragan Spaic, Nenad Markovic, Aleksandar Corac.

**Validation:** Zoran Bukumiric, Aleksandra Ilic, Dragana Srebro, Sasa Milicevic, Aleksandar Corac.

**Visualization:** Zoran Bukumiric, Dragan Spaic, Nenad Markovic.

**Writing – original draft:** Zoran Bukumiric.

**Writing – review & editing:** Zoran Bukumiric, Aleksandra Ilic, Mirjana Pajcin, Dragana Srebro, Sasa Milicevic, Dragan Spaic, Nenad Markovic, Aleksandar Corac.

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
