## [Decision Letter · Decision Letter 0]

26 Aug 2021

PONE-D-21-22424

Effects of problem-based learning modules within blended learning courses in medical statistics - randomized controlled pilot study

PLOS ONE

Dear Dr. Bukumirić,

Thank you for submitting your manuscript to PLOS ONE. After careful consideration, we feel that it has merit but does not fully meet PLOS ONE’s publication criteria as it currently stands. Therefore, we invite you to submit a revised version of the manuscript that addresses the points raised during the review process.

While both reviewers recognized the value and contribution of your study, both noted significant issues with the manuscript, in particular around reporting of study's methodology. Take a detailed look at the provided comments, I hope you find the feedback useful and reviewers' comments constructive. I'm looking forward to reading the revised manuscript.

We look forward to receiving your revised manuscript.

Kind regards,

Vitomir Kovanović, Ph.D.

Academic Editor

PLOS ONE

Journal Requirements:

Reviewers' comments:

Reviewer's Responses to Questions

**Comments to the Author**

1. Is the manuscript technically sound, and do the data support the conclusions?

Reviewer #1: Partly

Reviewer #3: Partly

2. Has the statistical analysis been performed appropriately and rigorously? 

Reviewer #1: Yes

Reviewer #3: Yes

3. Have the authors made all data underlying the findings in their manuscript fully available?

Reviewer #1: No

Reviewer #3: No

4. Is the manuscript presented in an intelligible fashion and written in standard English?

Reviewer #1: Yes

Reviewer #3: No

5. Review Comments to the Author

Reviewer #1: Thank you for the opportunity to review your manuscript. I have provided some potential suggestions for edits. Of note, it may be helpful to work with someone regarding the syntax and diction--there were several grammatical errors and challenges with the flow that could make the manuscript more impactful.

INTRO

- Provide a nice summary of applicable research and outline specifically the application to medical statistics. Would suggest to explore if there are publications on medical statistics / epidemiology teaching practices broadly in the literature. It may be helpful to put this within the context of other work there, not necessarily just PBL related.

- Several grammatical errors (e.g., "there were no studies on problem-based blended learning method for teaching the medical statistics)

METHODS

- Love that this was an RCT, we don't get to see that often! The main question is about the fidelity of the work--how did you prevent students from sharing information or resources?

- I don't understand the statement "the students passed the exam in medical statistics during the study". Is that more of a result?

- Can you provide more information about the context of the course this activity was embedded within? Also to double-check, there are normally 53 students in the program correct? Or was this the number who consented to participate.

- May be helpful to reference Figure 1 earlier in the methods when discussing the steps that the students went through.

- A summary table of the assessment activities may also be useful. Could you also provide a sample, maybe in an appendix to know what these activities looked like, time dedicated to them, etc.

- I have difficulty following how there was significant difference between the two groups or how they were sufficiently separated for the purpose of the study. Can you please provide more details about how their experiences differed based on the random assignment.

- It is not clear what assessment activities are included and how the relates to the linear regression. More details are needed. How were the problem solving and medical statistics score measured? What are the total possible scores? Have they been used in other settings that showcase how well they measure those two abilities? Etc. Since these are essential for interpreting the results more details are needed about the reliability and validity of these results.

RESULTS

- Can you please clarify "the grade expected" since these are number values; for international audience this may be useful if they are used to letters, etc.

- "The exam was passed" in the table is also confusing. I'm not sure what this means, the number of people who passed? What qualifies as passing the exam and when is the exam administered?

- Do you have the correlation between the problem solving and medical statistics scores, it would be curious to see the extent of their relationship to one another.

- For Likert scales, it's generally inappropriate to summarize at average/median levels. Instead, it is preferred to illustrate the number of individuals who responded with each choice.

DISCUSSION

- More comparison of statistical versus "clinical" significance may be necessary. The mean scores differed by 1-2 points, is that truly significant beyond statistically?

- Unfortunately, more details are needed about the methodology to evaluate the discussion appropriately.

Reviewer #3: Dear authors,

Congratulations on the work developed and submitted to this journal, on the effects of problem-based learning (PBL) modules within blended learning (BL) courses in medical statistics, by running a randomized controlled trial. Your aim was to assess the efficacy of PBL modules implemented within the blended-learning courses in medical statistics through the knowledge outcomes and student satisfaction.

Considering the above mentioned as a head start for the review, below you will find some comments on each section that should be addressed.

- Abstract Section:

Whilst being overall well written, this particular section should be restructured in order to be more appealing for the reader. There is an excess of abbreviations which lead to a sense of confusion. Additionally, it should mention briefly the future steps of this project.

- Introduction Section:

This particular section is underexplored. Some questions that should guide your thoughts while adding some important information are the reason why PBL methods can be important on the subject of medical statistics and the particular subset of skills that a medical student is supposed to develop from this curricula. Additionally, once you set out to explore knowledge outcomes and student satisfaction, there are previously conducted studies that explore these variables with future medical skills, so what is really the importance of student satisfaction on continuing learning profiles, e.g.? The last paragraph should be fully restructured, as it is one of the most important ones for the reader, since it is the last one before the next Materials and Methods section. I would advise to begin with the aim of the study and only then advance to the design made. Also, some english grammar errors were found, which are described below.

Line 37 - Where written "In the past decade, there are increasing (...)" change for "In the past decade, there is an increasing (...)"

Line 49 - Where written "During learning process the students (...)" change for "During the learning process, students (...)"

Line 50 - Where written "They" change for "The students" or "Students"

Paragraph 57 - Where written "To the best of our knowledge, there were no studies on problem-based blended learning method for teaching the medical statistics." change for "To the best of our knowledge, there are no previous studies on the problem-based blended learning method for medical statistics teaching."

- Materials and Methods Section:

First and foremost, once being a randomized controlled trial, it would be optimal to have it registered at the clinicaltrials.gov platform. Further on, the authors mention that an informed consent was obtained, however there is no reference on how. It is my opinion that these paragraphs 73/74/75 should add the environment in which the consent was obtained, to avoid any coercion interpretations. Additionally, you mention an anonymous online questionnaire for the assessment of the student satisfaction of the hPBL group. Was this preceded by a mandatory informed consent? How was the data managed and stored? Who had access to it? Were an IP addresses collected?

Overall, this section needs more work, mainly at the description of the method itself. I would a recomend a major revision of the english used, as well as the review of the duplicated information. Some sentences are in reality avoidable, because they are the continuation of the last.

Line 76 - Where written "The study began on the 1st of October, 2019 (...)" change for "The study began on October 1st, 2019 (...)

Line 77 - Where written "was completed at the end of school year (30.09.2020)" change for "was completed at the end of school year (September 30th, 2020)."

Lines 84/85 - Where written "Ministry of education, science and technological development of the Republic of Serbia" change for "Ministry of Education, Science and Technological Development of the Republic of Serbia"

Paragraph 86 - Should be rewritten, since there is repeated information.

Lines 106/107 - Rewrite the following sentence "All the necessary information is given to students during the lectures and exercises in the materials in the blended- learning course.". Suggestion: "All the necessary information was given to students during the lectures, while the exercises and materials where given at the blended-learning course."

Lines 126/127/128 - Where written "Anonymous online questionnaire with the five-point Likert scale (1 point- low satisfaction, 5 points- high satisfaction) was used to assess the satisfaction of the students in the hPBL group with the PBL modules." change for "An anonymous online questionnaire with the five-point Likert scale (1 point- low satisfaction, 5 points- high satisfaction) was used to assess the satisfaction of the students in the hPBL group with the PBL modules.".

- Results Section:

This section is well presented and clear. I would suggest that in line 139, where written "There were no differences between (...)", the authors change for "There were no statistically significant differences between (...)".

- Discussion Section:

The discussion section is well explored and written.

- Conclusions Section:

The conclusions are both supported by the results obtained in the present study, as well as previous studies.

This article has some flaws as pointed above, for each section. A general review of the english grammar and spell check should be performed, additionally to addressing the point mentioned.

Best regards,

6. PLOS authors have the option to publish the peer review history of their article (what does this mean?). If published, this will include your full peer review and any attached files.

Reviewer #1: No

Reviewer #3: No

---

## [Author Response · Author response to Decision Letter 0]

4 Oct 2021

Response to reviewer 1:

Dear Madame/Sir,

Thank you for your time and assistance. We appreciate very much your professional advice and accepted comments and suggestions in the text. 

We accepted all suggestions in Abstract, Introduction, Methods, Results, and Discussion section. 

Also, we added the recommended references. 

Comments to the authors: 

Thank you for the opportunity to review your manuscript. I have provided some potential suggestions for edits. Of note, it may be helpful to work with someone regarding the syntax and diction--there were several grammatical errors and challenges with the flow that could make the manuscript more impactful.

INTRO

- Provide a nice summary of applicable research and outline specifically the application to medical statistics. Would suggest to explore if there are publications on medical statistics / epidemiology teaching practices broadly in the literature. It may be helpful to put this within the context of other work there, not necessarily just PBL related.

Thank you for your comment. We added the paragraph on the existing teaching practices in the introduction section. 

‘Medical doctors can be exceptional in their fields even if they do not know medical statistics, but they will be better if they do [1]. The study of Swift et al [2] showed that medical doctors considered medical statistics useful for “accessing clinical guidelines and evidence summaries, explaining risk levels to patients, assessing medical marketing and advertising material, interpreting the results of a screening test, reading research publications for general professional interest, and using research publications to explore non-standard treatment and management options.” Future physicians also thought that there was a need for a practical application of knowledge in medical statistics, not only its’ theoretical basis [3]. Lack of knowledge in medical statistics can lead to misinterpretation of clinical findings [4]. Statistical softwares, widely available now, enable an easy and comfortable analysis, but mistakes can be made when choosing the appropriate statistical test or assumptions for its' application [5]. Medical students state that learning medical statistics through real life problems and the process of drawing conclusions can be more productive than traditional learning and knowledge assessment [6].' 

1. Trajkovic G. “Introduction to medical statistics”. Lectures. Faculty of Medicine, University of Pristina in Kosovska Mitrovica. 2015. 

2. Swift L, Miles S, Price GM, Shepstone L, Leinster SJ. Do doctors need statistics? Doctors’ use of and attitudes to probability and statistics. Stat Med. 2009;28: 1969–1981. doi:10.1002/sim.3608

3. MacDougall M, Cameron HS, Maxwell SRJ. Medical graduate views on statistical learning needs for clinical practice: A comprehensive survey. BMC Med Educ. 2019;20. doi:10.1186/s12909-019-1842-1

4. Windish DM, Huot SJ, Green ML. Medicine residents’ understanding of the biostatistics and results in the medical literature. J Am Med Assoc. 2007;298: 1010–1022. doi:10.1001/jama.298.9.1010

5. Masic I, Jankovic SM, Begic E. PhD Students and the Most Frequent Mistakes during Data Interpretation by Statistical Analysis Software. Studies in Health Technology and Informatics. IOS Press; 2019. pp. 105–109. doi:10.3233/SHTI190028

6. Astin J, Jenkins T, Moore L. Medical students’ perspective on the teaching of medical statistics in the undergraduate medical curriculum. Stat Med. 2002;21: 1003–1006. doi:10.1002/sim.1132

- Several grammatical errors (e.g., "there were no studies on problem-based blended learning method for teaching the medical statistics)

Thank you for your comment. We consulted the native English speaker to check the grammar and spelling after the revision. 

METHODS

- Love that this was an RCT, we don't get to see that often! The main question is about the fidelity of the work--how did you prevent students from sharing information or resources?

Thank you for your comment. The administrator of the Moodle platform checked if the students assessed the PBLs from the same IP address, which was true for our students. We could not control if the students from PBL group shared the information on modules with the students from BL group. We expect that there were no such cases, or were only a few and that these cases did not affect the differences in scores. This was a pilot study and its’ results will be used in further research. We added this in the limitation section: 

‘Another potential limitation of the study, we could not control for, was that the students from the hPBL group could show the modules to the students in BL group. This risk can be minimized through the inclusion of the larger number of participants in the study and through the development of a larger number of the problems for students to solve.’ 

- I don't understand the statement "the students passed the exam in medical statistics during the study". Is that more of a result?

Thank you for your comment. Our study lasted for 12 months, during which it is expected from all the students enrolled in any courses to pass the exams. The students pass the exam if they complete all the necessary course activities and if they obtain 51 or more points (out of the 100). We deleted this sentence, and provided the detailed explanation in the Materials and Methods section. 

- Can you provide more information about the context of the course this activity was embedded within? 

Thank you for your comment. We added the description in the Materials and methods section. 

‘Blended learning course in medical statistics and informatics is based on a Moodle platform and contains 15 classes of theoretical lectures, 30 classes of practical exercises and 15 classes of other type, such as online readings or seminars. Total of 70% of the program of this course is comprised of medical statistics and this part of the course contains units on data types, descriptive statistics, confidence interval, probability and probability distributions, hypotheses testing, correlation and linear regression. Practical exercises are done using the statistical software Easy R (EZR) [18]. Students from both groups in our study had access to identical course activities (lectures and exercises), except for the access to the problem-based modules that were available only the students from hPBL group (Table 1). During the course, students receive grades for all existing activities: lectures, exercises, colloquium, seminars, solving problems and final test. Students can see all the points for each activity any time during the course. The maximal number of points is 100 (70 for statistics and 30 for informatics). Students need to obtain the minimum of 51 points to pass the course. Based on the total number of points (51-100) the passing grades students can receive vary from 6 to 10.

Table 1 – Activities during Medical statistics and informatics course 

Timeline hPBL BL

Weekly Lectures Lectures

Weekly Practical exercises done using the statistical software Practical exercises done using the statistical software 

Weekly Independent students’ assignments (interactive online lectures, Moodle) Independent students’ assignments (interactive online lectures, Moodle)

Weekly Problem-based learning module (Moodle) –

During the course Seminars Seminars 

During the course Colloquium Colloquium 

At the end of the course Problem solving Problem solving 

At the end of the course Final test Final test

Our study examined only the outcomes of medical statistics (70% of the course): problem solving score (5 problems with maximum score of the total of 25 points) and total medical statistics score (theoretical knowledge score, practical exercises score, problem solving score, independent students’ assignments score, seminars and colloquium; the maximal total medical statistics score was 70 points).‘ 

Also to double-check, there are normally 53 students in the program correct? Or was this the number who consented to participate.

Thank you for your comment. There were total of 62 students in the third year of medical school and all of them gave the consent for participation in the study (30 hPBL group and 32 in BL group), but the 53 students completed all of the course activities and were included in the analysis. We considered this to be acceptable, without any influence on the results, as the drop- out from the study was less than 15%, considered acceptable for the longitudinal studies (https://www.cebm.ox.ac.uk/resources/levels-of-evidence/oxford-centre-for-evidence-based-medicine-levels-of-evidence-march-2009).

We clarified this in the abstract and in the Material and methods section: 

Abstract 

‘The pilot study was designed as a randomized controlled trial that included 53 medical students who completed all course activities.’

Material and methods

‘The final analysis included 53 students who had completed all course activities out of 62 students who had been initially included in the study.’

- May be helpful to reference Figure 1 earlier in the methods when discussing the steps that the students went through.

Thank you for noticing. We moved the figure 1 earlier in the methods section. 

PBL modules were created based on the structure of the steps in the statistical analysis (Figure 1).

 

- A summary table of the assessment activities may also be useful. Could you also provide a sample, maybe in an appendix to know what these activities looked like, time dedicated to them, etc.

- I have difficulty following how there was significant difference between the two groups or how they were sufficiently separated for the purpose of the study. Can you please provide more details about how their experiences differed based on the random assignment

Thank you for your comments. We provided the detailed description of all the activities in the Material and methods section. 

‘Blended learning course in medical statistics and informatics is based on a Moodle platform and contains 15 classes of theoretical lectures, 30 classes of practical exercises and 15 classes of other type, such as online readings or seminars. Total of 70% of the program of this course is comprised of medical statistics and this part of the course contains units on data types, descriptive statistics, confidence interval, probability and probability distributions, hypotheses testing, correlation and linear regression. Practical exercises are done using the statistical software Easy R (EZR) [18]. Students from both groups in our study had access to identical course activities (lectures and exercises), except for the access to the problem-based modules that were available only the students from hPBL group (Table 1). During the course, students receive grades for all existing activities: lectures, exercises, colloquium, seminars, solving problems and final test. Students can see all the points for each activity any time during the course. The maximal number of points is 100 (70 for statistics and 30 for informatics). Students need to obtain the minimum of 51 points to pass the course. Based on the total number of points (51-100) the passing grades students can receive vary from 6 to 10.

Table 1 – Activities during Medical statistics and informatics course 

Timeline hPBL BL

Weekly Lectures Lectures

Weekly Practical exercises done using the statistical software Practical exercises done using the statistical software 

Weekly Independent students’ assignments (interactive online lectures, Moodle) Independent students’ assignments (interactive online lectures, Moodle)

Weekly Problem-based learning module (Moodle) –

During the course Seminars Seminars 

During the course Colloquium Colloquium 

At the end of the course Problem solving Problem solving 

At the end of the course Final test Final test

Our study examined only the outcomes of medical statistics (70% of the course): problem solving score (5 problems with maximum score of the total of 25 points) and total medical statistics score (theoretical knowledge score, practical exercises score, problem solving score, independent students’ assignments score, seminars and colloquium; the maximal total medical statistics score was 70 points).‘ 

- It is not clear what assessment activities are included and how the relates to the linear regression. More details are needed. How were the problem solving and medical statistics score measured? What are the total possible scores? Have they been used in other settings that showcase how well they measure those two abilities? Etc. Since these are essential for interpreting the results more details are needed about the reliability and validity of these results.

Thank you for the comment. We added the explanation to the Materials and methods section. 

‘Our study examined only the outcomes of medical statistics (70% of the course): problem solving score (5 problems with maximum score of the total of 25 points) and total medical statistics score (theoretical knowledge score, practical exercises score, problem solving score, independent students’ assignments score, seminars and colloquium; the maximal total medical statistics score was 70 points).’

RESULTS

- Can you please clarify "the grade expected" since these are number values; for international audience this may be useful if they are used to letters, etc.

Thank you for noticing this. The ‘grade expected’ referred to the grade that student stated that is expected to get in Medical statistics and informatics. Students were asked at the beginning of the course which grade they are expecting to get at the end of it. The passing grades at our University vary from 6 to 10, and the grade 5 means that the student failed the exam. The passing grades are computed based on the total points obtained during the course (0-100). We provided the explanation in the methods section: 

‘During the course, students receive grades for all existing activities: lectures, exercises, colloquium, seminars, solving problems and final test. Students can see all the points for each activity any time during the course. The maximal number of points is 100 (70 for statistics and 30 for informatics). Students need to obtain the minimum of 51 points to pass the course. Based on the total number of points (51-100) the passing grades students can receive vary from 6 to 10.’

- "The exam was passed" in the table is also confusing. I'm not sure what this means, the number of people who passed? What qualifies as passing the exam and when is the exam administered?

Thank you for the comment. We understand that it was not clearly described in the methods and in the results section. We added the explanation of the minimum of activities student needs to complete in order to complete the course and rephrased the ‘passing the exam’ to successfully completing the course in medical statistics in Table 2. 

- Do you have the correlation between the problem solving and medical statistics scores, it would be curious to see the extent of their relationship to one another.

Thank you for the comment. The problem solving score is a part of total medical statistics score, which is why we did not examine the correlation between them. We described this in the material and methods section: 

‘total medical statistics score (theoretical knowledge score, practical exercises score, problem solving score, independent students’ assignments score, seminars and colloquium; the maximal total medical statistics score was 70 points).’

 

- For Likert scales, it's generally inappropriate to summarize at average/median levels. Instead, it is preferred to illustrate the number of individuals who responded with each choice.

Thank you for your comment. The responses in our questionnaire did not have the names of the categories, but were numerical (1-5), which is why we considered that it is adequate to show them through median and range. Additionally, as seen in table, the minimum in our study was 3. However, in accordance with your comments, we added the column in table 5 with the frequency of the most frequent response. 

Table 6. Students’ attitudes towards the implemented problem solving modules 

Question median (range) n (%)

highest satisfaction 

Problems are adequate for learning medical statistics 5 (3 – 5) 19 (73.1%)

Contents and structure of problem-based modules is interesting 4.5 (3 – 5) 13 (50.0%)

I like this modality of solving the actual statistical problems 5 (3 – 5) 19 (73.1%)

Step- by- step approach within the module is useful for learning medical statistics 5 (3 – 5) 20 (76.9%)

Problem based modules helped me to learn the steps for resolving actual statistical problems 5 (3 – 5) 21 (80.8%)

Problem-based modules helped me understand medical statistics 5 (3 – 5) 20 (76.9%)

DISCUSSION

- More comparison of statistical versus "clinical" significance may be necessary. The mean scores differed by 1-2 points, is that truly significant beyond statistically? 

Along with the statistical significance, we calculated the effect size, as an approximation to the 'clinical' significance. In our study, the effect size was moderate (0.69 for problem solving score and 0.55 for total medical statistics score). The larger effect size is associated with the power of the statistical test used correctly reject a false null hypothesis, or probability that the test will identify a treatment effect if one really exists. In our study, the solving of PBL modules increased not only the problem solving score, but also the total medical statistics score. This can lead to the increase in the capability of the physicians to apply medical statistics in their daily work. We consider that this will be confirmed in the future research on the larger sample. Additionaly, the differences between the groups remained significant in the multivariate models as well. 

- Unfortunately, more details are needed about the methodology to evaluate the discussion appropriately.

We hope that the new paragraphs added in the introduction and materials and methods clarify the methodology and enable easier reading of the discussion section. 

################

Response to reviewer 3:

Dear Madame/Sir,

Thank you for your time and assistance. We appreciate very much your professional advice and accepted comments and suggestions in the text. 

We accepted all suggestions in Abstract, Introduction, Methods, Results, and Discussion section. 

Also, we added the recommended references. 

Comments to the authors: 

Dear authors,

Congratulations on the work developed and submitted to this journal, on the effects of problem-based learning (PBL) modules within blended learning (BL) courses in medical statistics, by running a randomized controlled trial. Your aim was to assess the efficacy of PBL modules implemented within the blended-learning courses in medical statistics through the knowledge outcomes and student satisfaction.

Considering the above mentioned as a head start for the review, below you will find some comments on each section that should be addressed.

- Abstract Section:

Whilst being overall well written, this particular section should be restructured in order to be more appealing for the reader. There is an excess of abbreviations which lead to a sense of confusion. 

Additionally, it should mention briefly the future steps of this project.

Thank you for your comment. We rewrote the abstract in order to be more understandable. We also added sentence on the future research: 

'The authors of this study are planning to create PBL modules for advanced courses in medical statistics and to conduct this study on other universities with a more representative study sample, with the aim to overcome the limitations of the existing study and confirm its results.’

 

- Introduction Section:

This particular section is underexplored. Some questions that should guide your thoughts while adding some important information are the reason why PBL methods can be important on the subject of medical statistics and the particular subset of skills that a medical student is supposed to develop from this curricula. Additionally, once you set out to explore knowledge outcomes and student satisfaction, there are previously conducted studies that explore these variables with future medical skills, so what is really the importance of student satisfaction on continuing learning profiles, e.g.? 

Thank you for your valuable comment. We added the paragraph in the introduction section. 

‘Medical doctors can be exceptional in their fields even if they do not know medical statistics, but they will be better if they do [1]. The study of Swift et al [2] showed that medical doctors considered medical statistics useful for “accessing clinical guidelines and evidence summaries, explaining risk levels to patients, assessing medical marketing and advertising material, interpreting the results of a screening test, reading research publications for general professional interest, and using research publications to explore non-standard treatment and management options.” Future physicians also thought that there was a need for a practical application of knowledge in medical statistics, not only its’ theoretical basis [3]. Lack of knowledge in medical statistics can lead to misinterpretation of clinical findings [4]. Statistical softwares, widely available now, enable an easy and comfortable analysis, but mistakes can be made when choosing the appropriate statistical test or assumptions for its' application [5]. Medical students state that learning medical statistics through real life problems and the process of drawing conclusions can be more productive than traditional learning and knowledge assessment [6].' 

1. Trajkovic G. “Introduction to medical statistics”. Lectures. Faculty of Medicine, University of Pristina in Kosovska Mitrovica. 2015. 

2. Swift L, Miles S, Price GM, Shepstone L, Leinster SJ. Do doctors need statistics? Doctors’ use of and attitudes to probability and statistics. Stat Med. 2009;28: 1969–1981. doi:10.1002/sim.3608

3. MacDougall M, Cameron HS, Maxwell SRJ. Medical graduate views on statistical learning needs for clinical practice: A comprehensive survey. BMC Med Educ. 2019;20. doi:10.1186/s12909-019-1842-1

4. Windish DM, Huot SJ, Green ML. Medicine residents’ understanding of the biostatistics and results in the medical literature. J Am Med Assoc. 2007;298: 1010–1022. doi:10.1001/jama.298.9.1010

5. Masic I, Jankovic SM, Begic E. PhD Students and the Most Frequent Mistakes during Data Interpretation by Statistical Analysis Software. Studies in Health Technology and Informatics. IOS Press; 2019. pp. 105–109. doi:10.3233/SHTI190028

6. Astin J, Jenkins T, Moore L. Medical students’ perspective on the teaching of medical statistics in the undergraduate medical curriculum. Stat Med. 2002;21: 1003–1006. doi:10.1002/sim.1132

 

The last paragraph should be fully restructured, as it is one of the most important ones for the reader, since it is the last one before the next Materials and Methods section. I would advise to begin with the aim of the study and only then advance to the design made. Also, some english grammar errors were found, which are described below.

Thank you for your comment. We rephrased the paragraph. 

‘The aim of this study was to evaluate the effectiveness of implemented problem-based modules within blended learning courses in medical statistics through the outcomes of knowledge and student satisfaction. We created problem-based modules in medical statistics, based on actual problems which contained all of the steps in statistical analysis (defining the problem, choosing and applying adequate statistical tests, interpreting the results and drawing conclusions) and implemented them within the blended learning course.’ 

Line 37 - Where written "In the past decade, there are increasing (...)" change for "In the past decade, there is an increasing (...)"

Thank you for your comment, we corrected this. 

Line 49 - Where written "During learning process the students (...)" change for "During the learning process, students (...)"

Thank you for your comment, we corrected this.

Line 50 - Where written "They" change for "The students" or "Students"

Thank you for your comment, we corrected this.

Paragraph 57 - Where written "To the best of our knowledge, there were no studies on problem-based blended learning method for teaching the medical statistics." change for "To the best of our knowledge, there are no previous studies on the problem-based blended learning method for medical statistics teaching."

Thank you for your comment, we corrected this.

- Materials and Methods Section:

First and foremost, once being a randomized controlled trial, it would be optimal to have it registered at the clinicaltrials.gov platform. 

Thank you for your comment. However, our study is a randomized controlled pilot study, and the intervention and outcomes are associated with the education of medical students. As we did not examine any clinical outcome, we did not think that the study fulfills the criteria for registration at the clinicaltrials.gov. 

Further on, the authors mention that an informed consent was obtained, however there is no reference on how. It is my opinion that these paragraphs 73/74/75 should add the environment in which the consent was obtained, to avoid any coercion interpretations. 

Thank you for noticing this. We added the section regarding the ethics in our study. 

'Ethics Statement

The study was approved by the Ethical Committee of the Faculty of Medicine, University of Pristina, Kosovska Mitrovica (No. 09-3171). During the first week of the course (the first week of the semester), before the randomization in the groups was performed, students received written and oral explanations of the study, the processes and aims, the modalities of data gathering and data analysis. Students were explained that all the information gathered would be anonymous, that the participation was voluntary and that they could dropout of the study at any point. After this, the students gave an oral consent for their participation in the study, which was then verified in their records. A questionnaire on satisfaction was filled in after the course had been completed, as an online anonymous and non-obligatory questionnaire. All the data on the course outcomes and the data from the questionnaire on the students’ satisfaction were gathered by the administrator of the Moodle platform who was the only one with the access to the complete database. The authors of the study only had access to the anonymized database that is provided with the manuscript.’

Additionally, you mention an anonymous online questionnaire for the assessment of the student satisfaction of the hPBL group. Was this preceded by a mandatory informed consent? 

Thank you for your comment. The questioonare was not obligatory and the procedure is described in the ethical statement section: 

‘A questionnaire on satisfaction was filled in after the course had been completed, as an online anonymous and non-obligatory questionnaire.’

How was the data managed and stored? Who had access to it? Were an IP addresses collected?

Thank you for the comment. We did not collect the IP addresses. The explanation was given in the Ethical statement: 

‘All the data on the course outcomes and the data from the questionnaire on the students’ satisfaction were gathered by the administrator of the Moodle platform who was the only one with the access to the complete database. The authors of the study only had access to the anonymized database that is provided with the manuscript.’

Overall, this section needs more work, mainly at the description of the method itself. I would a recomend a major revision of the english used, as well as the review of the duplicated information. Some sentences are in reality avoidable, because they are the continuation of the last.

Line 76 - Where written "The study began on the 1st of October, 2019 (...)" change for "The study began on October 1st, 2019 (...)

Thank you for your comment, we corrected this.

Line 77 - Where written "was completed at the end of school year (30.09.2020)" change for "was completed at the end of school year (September 30th, 2020)."

Thank you for your comment, we corrected this.

Lines 84/85 - Where written "Ministry of education, science and technological development of the Republic of Serbia" change for "Ministry of Education, Science and Technological Development of the Republic of Serbia"

Thank you for your comment, we corrected this.

 

Paragraph 86 - Should be rewritten, since there is repeated information.

Thank you for your comment, we rephrased the paragraph. 

‘PBL modules were created based on the structure of the steps in statistical analysis (Figure 1). Statistical analysis of the research problem is based on the multiple successive steps which include the following: the definition of the problem and the research question, recognition of the data type, sample type and hypothesis, selection of the adequate statistical test, application of the test, interpretation of the results and conclusion related to the description of the data, statistical conclusion and implications of the results. The PBL modules use guiding questions following the steps of statistical analysis. The guiding questions consisted of: interactive multiple choice or open-ended questions and followed a similar principle to the one Brown et al. applied [19]. Guiding questions changed within each step, based on the type of the statistical problem, number of variables and the sample (examples can be seen online following the link provided in the text bellow). Students can understand the necessary components for statistical reasoning by answering these questions and learn how to solve the problem. ’

Lines 106/107 - Rewrite the following sentence "All the necessary information is given to students during the lectures and exercises in the materials in the blended- learning course.". Suggestion: "All the necessary information was given to students during the lectures, while the exercises and materials were given at the blended-learning course."

Thank you for your comment, we corrected this.

Lines 126/127/128 - Where written "Anonymous online questionnaire with the five-point Likert scale (1 point- low satisfaction, 5 points- high satisfaction) was used to assess the satisfaction of the students in the hPBL group with the PBL modules." change for "An anonymous online questionnaire with the five-point Likert scale (1 point- low satisfaction, 5 points- high satisfaction) was used to assess the satisfaction of the students in the hPBL group with the PBL modules.".

Thank you for your comment, we corrected this.

- Results Section:

This section is well presented and clear. 

I would suggest that in line 139, where written "There were no differences between (...)", the authors change for "There were no statistically significant differences between (...)".

Thank you for your comment, we corrected this.

- Discussion Section:

The discussion section is well explored and written.

Thank you for the comment.

- Conclusions Section:

The conclusions are both supported by the results obtained in the present study, as well as previous studies.

This article has some flaws as pointed above, for each section. A general review of the english grammar and spell check should be performed, additionally to addressing the point mentioned.

Thank you for the comment. We also consulted the native English speaker for the revised version of the manuscript.

---

## [Decision Letter · Decision Letter 1]

11 Jan 2022

Effects of problem-based learning modules within blended learning courses in medical statistics – randomized controlled pilot study

PONE-D-21-22424R1

Dear Dr. Bukumiric,

We’re pleased to inform you that your manuscript has been judged scientifically suitable for publication and will be formally accepted for publication once it meets all outstanding technical requirements.

Kind regards,

Gwo-Jen Hwang

Academic Editor

PLOS ONE

Additional Editor Comments (optional):

Reviewers' comments:

Reviewer's Responses to Questions

**Comments to the Author**

1. If the authors have adequately addressed your comments raised in a previous round of review and you feel that this manuscript is now acceptable for publication, you may indicate that here to bypass the “Comments to the Author” section, enter your conflict of interest statement in the “Confidential to Editor” section, and submit your "Accept" recommendation.

Reviewer #1: All comments have been addressed

2. Is the manuscript technically sound, and do the data support the conclusions?

Reviewer #1: Yes

3. Has the statistical analysis been performed appropriately and rigorously? 

Reviewer #1: Yes

4. Have the authors made all data underlying the findings in their manuscript fully available?

Reviewer #1: Yes

5. Is the manuscript presented in an intelligible fashion and written in standard English?

Reviewer #1: Yes

6. Review Comments to the Author

Reviewer #1: Thank you for the response to the comments--I feel they have greatly enhanced the publication.

---

## [Editor Report · Acceptance letter]

14 Jan 2022

PONE-D-21-22424R1 

Effects of problem-based learning modules within blended learning courses in medical statistics – a randomized controlled pilot study 

Dear Dr. Bukumiric:

I'm pleased to inform you that your manuscript has been deemed suitable for publication in PLOS ONE. Congratulations! Your manuscript is now with our production department. 

Kind regards, 

on behalf of

Dr. Gwo-Jen Hwang 

Academic Editor

PLOS ONE